# Hybrid Operating Room System for the Treatment of Thoracic and Abdominal Aortic Aneurysms: Evaluation of the Radiation Dose Received by Patients

**DOI:** 10.3390/diagnostics10100846

**Published:** 2020-10-19

**Authors:** Yoshihiro Haga, Koichi Chida, Masahiro Sota, Yuji Kaga, Mitsuya Abe, Yohei Inaba, Masatoshi Suzuki, Taiichiro Meguro, Masayuki Zuguchi

**Affiliations:** 1Course of Radiological Technology, Health Sciences, Tohoku University Graduate School of Medicine, 2-1 Seiryo, Aoba, Sendai, Miyagi 980-8575, Japan; yoshihiro.hg@gmail.com (Y.H.); masahiro.st63728@gmail.com (M.S.); inaba@med.tohoku.ac.jp (Y.I.); msuzuki@irides.tohoku.ac.jp (M.S.); qqrm6wq9k@arrow.ocn.ne.jp (M.Z.); 2Department of Radiology, Sendai Kousei Hospital, 4-15 Hirosemachi, Aoba, Sendai 980-0873, Japan; kaga@sendai-kousei-hospital.jp (Y.K.); mitsuya@sendai-kousei-hospital.jp (M.A.); 3Department of Radiation Disaster Medicine, International Research Institute of Disaster Science, Tohoku University, 468-1 Aramaki Aza-Aoba, Aoba, Sendai, Miyagi 980-0845, Japan; 4Department of Cardiovascular Medicine, Sendai Kousei Hospital, 4-15 Hirose-machi, Aoba, Sendai 980-0873, Japan; meguro-t@sendai-kousei-hospital.jp

**Keywords:** fluoroscopy, interventional radiology (IVR), stent graft, aneurysm, endovascular treatment, X-ray examination, radiation safety, radiation dose, disaster medicine

## Abstract

In recent years, endovascular treatment of aortic aneurysms has attracted considerable attention as a promising alternative to traditional surgery. Hybrid operating room systems (HORSs) are increasingly being used to perform endovascular procedures. The clinical benefits of endovascular treatments using HORSs are very clear, and these procedures are increasing in number. In procedures such as thoracic endovascular aortic repair (TEVAR) and endovascular aortic repair (EVAR), wires and catheters are used to deliver and deploy the stent graft in the thoracic/abdominal aorta under fluoroscopic control, including DSA. Thus, the radiation dose to the patient is an important issue. We determined radiation dose indicators (the dose–area product (DAP) and air karma (AK) parameters) associated with endovascular treatments (EVAR and TEVAR) using a HORS. As a result, the mean ± standard deviation (SD) DAPs of TEVAR and EVAR were 323.7 ± 161.0 and 371.3 ± 186.0 Gy × cm^2^, respectively. The mean ± SD AKs of TEVAR and EVAR were 0.92 ± 0.44 and 1.11 ± 0.54 Gy, respectively. The mean ± SD fluoroscopy times of TEVAR and EVAR were 13.4 ± 7.1 and 23.2 ± 11.7 min, respectively. Patient radiation dose results in this study of endovascular treatments using HORSs showed no deterministic radiation effects, such as skin injuries. However, radiation exposure during TEVAR and EVAR cannot be ignored. The radiation dose should be evaluated in HORSs during endovascular treatments. Reducing/optimizing the radiation dose to the patient in HORSs is important.

## 1. Introduction

In recent years, endovascular treatment of aortic aneurysms has attracted considerable attention as a promising alternative to traditional surgery [1]. Endovascular treatment reduces the risk of complications and shortens hospital stays [2]. In thoracic endovascular aortic repair (TEVAR) and endovascular aortic repair (EVAR), wires and catheters are used to deliver stent grafts to and deploy the grafts in the thoracic/abdominal aorta under fluoroscopic control (including digital subtraction angiography (DSA)). The procedures are often complex, associated with a long fluoroscopy time (FT) and many cine acquisitions, thus delivering high radiation doses to both the interventional radiology (IVR) staff [3,4,5] and patient [6]. The quality control of fluoroscopic X-ray devices [7,8] and management of exposure doses are important issues. Appropriate stent graft placement requires very precise manipulation. Mobile C-arms can often overheat, and exhibit image degradation [9]. Thus, the X-ray fluoroscopic device must feature both a large capacity and high X-ray output. Hybrid operating room systems (HORSs) are increasingly being used to perform endovascular procedures. A HORS combines state-of-the-art imaging with the use of a high-capacity angiographic system, affording optimal patient care. The fact that the operating room is sterile enables vascular surgeons to combine complex endovascular procedures with open surgery [10]. Major clinical benefits are already apparent, and HORSs are increasing in popularity.

The radiation doses delivered to small groups of patients undergoing endovascular treatment of thoracic and abdominal aortic aneurysms were quantified previously. We here attempted to optimize the radiation doses delivered during HORS-mediated treatment of patients with thoracic and abdominal aortic aneurysms. Direct methods of assessing radiation exposure [11,12,13,14] are cumbersome; thus, we used indirect methods (the dose–area product (DAP) and air karma (AK) parameters) to assess skin radiation doses [15,16,17,18,19,20,21,22,23,24,25,26,27,28]. We determined the DAP and AK radiation dose indicators associated with EVAR and TEVAR delivered via a HORS.

## 2. Materials and Methods

The present study was conducted in a HORS at Sendai Kousei Hospital (Japan). We evaluated the radiation dose indicators (DAP and AK) and related factors (i.e., FT) in 256 consecutive patients undergoing endovascular treatment from 2010 to 2012. The study was approved by the ethics committee of Sendai Kousei Hospital (Permission code: 30-19; 28 May 2018). All procedures were performed using a digital X-ray system (INFX-8000H, Toshiba, Otawara, Japan; Figure 1). We evaluated single-plane images obtained using a large field-of-view (12 × 16 in). Digital cine images were acquired at 10 frames/s during all procedures. Pulsed fluoroscopy (7.5 pulses/s) featuring flat panel detection (FPD) using an anti-scatter grid were also performed. During the X-ray procedures, including DSA, digital angiography (DA), and fluoroscopy, TEVAR was performed using a 40–50° left anterior oblique view, whereas EVAR was performed using various viewing angles. The X-ray source was placed approximately 100 cm from the FPD. The X-ray details and contrast conditions used during TEVAR and EVAR are shown in Table 1.

The DAP, AK, FT, and number of DA or DSA procedures performed were recorded for all patients. AK was measured at a point 15 cm toward the focal spot commencing at the isocenter of the C-arm type fluoroscope (the interventional reference point) in line with the standards of the International Electrotechnical Commission (IEC). The radiation dose indicators (DAP and AK) were calibrated by the manufacturer.

The data of the patients treated via TEVAR and EVAR are summarized in Table 2. The choice of a TEVAR stent graft (a total of 84 cases) was determined by reference to individual anatomies and included Zenith TX2 (34 cases; Cook, Japan), Tag (44 cases; Gore, Japan), and Valiant (6 cases; Medtronic, Japan) stents. The choice of an EVAR stent graft (in a total of 172 cases) was also determined by reference to individual anatomies and included Zenith (29 cases; Cook), Excluder (58 cases; Gore), Talent (7 cases; Medtronic), Endurant (53 cases; Medtronic), and Powerlink (25 cases; Cosmotec, Japan) stents.

### 2.1. Features of the HORS

The HORS is a hybrid device consisting of an angiographic X-ray device and an operating room (Figure 1). The HORS affords more advantages than do mobile C-arm systems. For example, the X-ray tube has a large capacity, the X-ray generator has a high output, and the FPD is large (12 × 16 in). Remote operation (control) of the X-ray equipment is possible. Such remote operation (control) reduces the exposure of radiology staff (such as radiographers) to radiation. Additionally, the operating table tilts. Surgery involving the brachiocephalic, common carotid, and subclavian arteries, which originate from the aortic arch, requires a head-down position to prevent a cerebral arterial air embolism. The HORS can easily be converted to an emergency surgery setting if complications develop during a procedure.

### 2.2. Statistics

Scatter plots were created and Pearson correlation coefficients between continuous variables were calculated. The Student’s *t*-test was used to compare data between the TEVAR and EVAR groups. Fisher’s exact test was employed to compare the male-to-female ratios. Statistical significance was defined as *p <* 0.05.

## 3. Results

### 3.1. Radiation Doses Associated with TEVAR and EVAR

The data are summarized in Table 3. The two groups did not differ significantly except in terms of age.

The mean ± standard deviation (SD) DAPs of TEVAR and EVAR were 323.7 ± 161.0 and 371.3 ± 186.0 Gy × cm^2^, respectively. The mean ± SD AKs of TEVAR and EVAR were 0.92 ± 0.44 and 1.11 ± 0.54 Gy, respectively. The DAP and AK of EVAR were approximately 1.2-fold higher than those of TEVAR. The mean ± SD FTs of TEVAR and EVAR were 13.4 ± 7.1 and 23.2 ± 11.7 min; the FT of TEVAR was typically somewhat shorter. The mean ± SD number of acquisitions (DAs and DSAs) for TEVAR and EVAR were 18.7 ± 9.8 and 29.8 ± 11.0 respectively. Figure 2 and Figure 3 show the correlations between the radiation dose and related factors in TEVAR or EVAR, respectively. Significant correlations were apparent between the AKs and FTs (TEVAR: *r* = 0.678, *p* < 0.001; EVAR: *r* = 0.616, *p* < 0.001). The correlation for TEVAR was superior to that for EVAR. More than 2 Gy was involved in 15 (6%) patients (2 TEVAR and 13 EVAR). Nevertheless, no radiation skin injury was observed in this study.

### 3.2. TEVAR and EVAR (Literature Review) [1,2,9,10,15,16,17,18,19,22,26,27,28]

Only limited data are available on the radiation doses associated with TEVAR and EVAR in the context of HORSs. To measure exposure during stent grafting, we considered DAP, AK, and FT. Table 4 shows the median (mean ± SD) DAP, AK, and FT for TEVAR (eight studies) delivered using a HORS. The FT ranged from 11.8 to 82.7 min (mean: 9.7–111.3 min). The FTs of two reports (Panuccio [15] and Mohapatra [17]) were longer, possibly because the cited authors treated many difficult cases and performed elaborate procedures (e.g., implantation of fenestrated stent grafts). The AKs for TEVAR ranged from 0.8 to 6.3 mGy, with 0.8 mGy (our study) being the lowest ever reported.

Table 5 shows the median (mean ± SD) DAP, AK, and FT for EVAR (16 studies) delivered using a HORS. The FT ranged from 13.1 to 89.0 min. The AK ranged from 0.56 to 1.26 Gy. Our FT and AK values were within these ranges.

Figure 4 shows the correlations between the radiation dose (AK and DAP) and FT during the endovascular treatment of aortic aneurysms in all reports that employed HORSs to this end. The scatter plots were constructed using mean values when possible and medians otherwise. With the exception of the EVAR for DAP and FT, correlations were evident between the radiation dose (DAP and AK) and FT. It is difficult to compare DAPs across reports, as the DAP differs according to the hospital and X-ray equipment employed. In particular, the DAPs of EVAR are influenced by changes in the field-of-view depending on the treatment applied.

## 4. Discussion

Radiation protection of patients and physicians in X-ray examination is important [29,30,31,32,33,34,35,36,37,38,39,40,41]. Radiation exposure in HORSs is of considerable interest. In the present study, we used clinical radiation-related data (DAP, AK, and FT) to define the exposure trends during stent grafting in HORSs. A total of 15 (6%) patients (2 TEVAR and 13 EVAR patients) were exposed to >2 Gy. However, no radiation skin injury was observed. Abdominal aortic operations were associated with higher radiation doses and FTs than thoracic aortic operations. The numbers of exposures, and their durations, were directly related to the radiation dose. Prolongation of fluoroscopy by 10 min in TEVAR or EVAR increased the AKs by approximately 0.78 or 0.72 Gy, respectively. Therefore, it is important to monitor the FT during both TEVAR and EVAR.

The dose levels in our study are shown in Table 4 and Table 5. Moreover, the FTs associated with AKs >2 Gy were approximately 20 min for TEVAR and 35 min for EVAR (Figure 4). These were significantly shorter than those associated with various IVRs. If these values are significantly exceeded, surgery may need to be re-planned. To allow DNA recovery following radiation injury, surgery can be performed in two or three stages. TEVAR was associated with lower FTs but higher radiation doses compared with EVAR. The oblique C-arm angulation is rather extreme when aortic neck vessels are viewed perpendicularly. In such circumstances, the radiation dose is increased, because the X-rays need to travel for longer distances through tissue (in other words, the kV/mA is increased to compensate for the lack of penetration in the tissue). Furthermore, difficult procedures, such as fenestrated stent graft implantation, tended to be associated with higher FTs and radiation doses.

The more DAs and DSAs that are employed, the higher the radiation doses. Thus, using fluoroscopy instead of DA and/or DSA can reduce the doses associated with TEVAR and EVAR.

Virtual 3D-CT pre-operative planning was very helpful when choosing an interventional strategy. The combination of 3D-CT and fluoroscopy, as is possible in a HORS, greatly facilitates surgery.

Endovascular treatment of aortic aneurysms performed in a HORS is thus safe and effective. Varu [22] found that EVAR in a HORS reduced the total operative time by >30 min, thus reducing/optimizing the radiation dose using HORS.

In our study, we provide suggested diagnostic reference levels (DRLs; ICRP—recommended; i.e., radiation doses) for TEVAR and EVAR performed in a HORS.

Generally, correlations between the number of acquisitions and radiation dose indicators were low (Figure 2 and Figure 3). Therefore, the number of acquisitions is not a useful parameter for radiation dose.

Other important considerations must be addressed in terms of reducing radiation exposure. One is image quality, deterioration of which may increase the risk of a high radiation dose. Image quality assessment is possible using phantoms. We always adjust image quality and exposure dose using a quality-control phantom. Furthermore, all IVR staff require periodic training in radiation safety, reducing doses to both patients and physicians.

Physician behavior can also reduce the radiation dose, such as the use of the shortest fluoroscopy time possible [42]. The ICRP [43] also recommends that physicians, ‘‘Keep beam-on time to an absolute minimum—the golden rule for control of dose to patient.’’

In summary, the clinical benefits of endovascular treatments using HORSs are very clear, and these procedures are increasing in number. In procedures such as EVAR and TEVAR, wires and catheters are used to deliver and deploy the stent graft in the thoracic/abdominal aorta under fluoroscopic control, including DSA. Thus, the radiation dose to the patient is an important issue. Thus, we determined radiation doses associated with endovascular treatments (EVAR and TEVAR) using HORS. Patient radiation dose results in this study of endovascular treatments using HORS showed no deterministic radiation effects, such as skin injuries. The radiation dose should be evaluated in HORS during endovascular treatments. Reducing/optimizing the radiation dose to the physician and patient in HORSs is important. Generally, the methods for reducing/optimizing the radiation dose in HORSs will be the same as those typical for interventional radiology X-ray systems.

## 5. Conclusions

We explored the radiation dose indicators (i.e., AK and DAP) delivered during TEVAR and EVAR performed in a HORS. We aimed to maximize patient safety. Radiation exposure during TEVAR and EVAR cannot be ignored. The FT, DAP, and AK during endovascular treatment of aortic aneurysm all increase with procedural complexity. In our study, some patients were exposed to >2 Gy (AK dose). An AK ≤2 Gy does not trigger a skin disorder, so the AK should optimally not exceed 2 Gy. All surgeons, radiologists, cardiologists, nurses, and radiographers must make efforts to minimize radiation exposure during TEVAR and EVAR. We show that the use of HORS during endovascular treatment of aortic aneurysms usefully reduces radiation exposure.

## Figures and Tables

**Figure 1 diagnostics-10-00846-f001:**
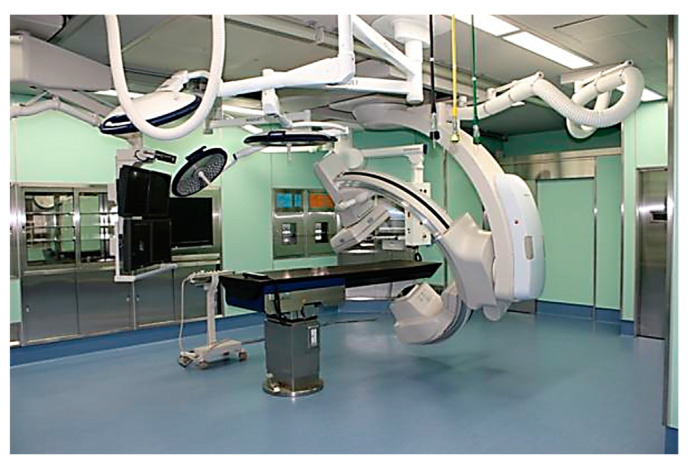
Photograph of the hybrid operating room system (HORS).

**Figure 2 diagnostics-10-00846-f002:**
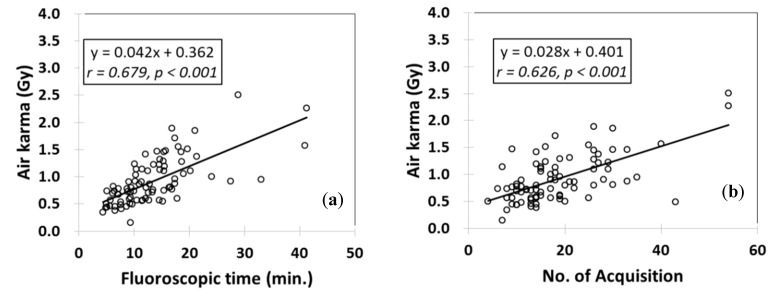
Relationships between the radiation doses during TEVAR and other factors. (**a**) The relationship between air karma (AK) and fluoroscopy time (FT). (**b**) The relationship between AK and the number of acquisitions (DA and DSA). (**c**) The relationship between DAP and FT. (**d**) The relationship between DAP and the number of acquisition (DA and DSA).

**Figure 3 diagnostics-10-00846-f003:**
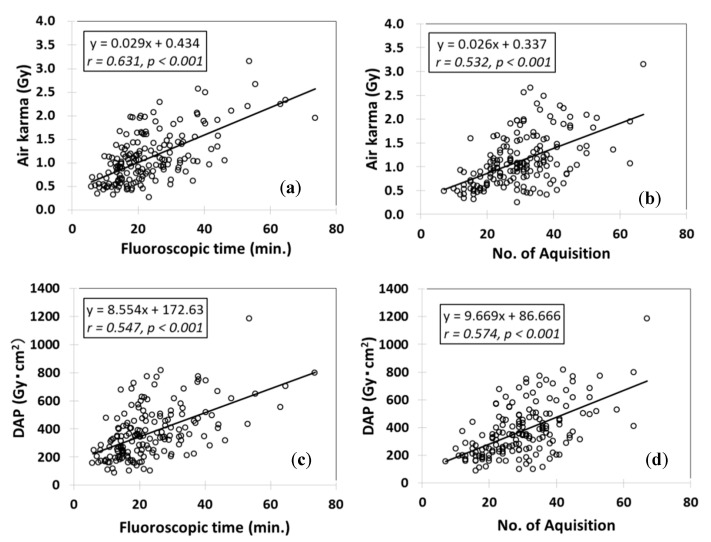
Relationships between the radiation doses during EVAR and other factors. (**a**) The relationship between air karma (AK) and fluoroscopy time (FT). (**b**) The relationship between AK and the number of acquisitions (DA and DSA). (**c**) The relationship between DAP and FT. (**d**) The relationship between DAP and the number of acquisition (DA and DSA).

**Figure 4 diagnostics-10-00846-f004:**
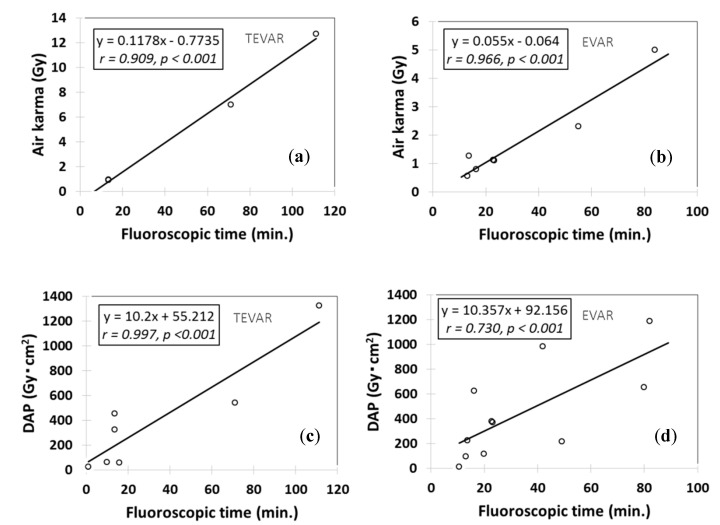
Relationships between the radiation doses (AKs and DAPs) and FT during endovascular treatment of aortic aneurysms in HORSs, as calculated from data from all relevant published reports. (**a**) The relationship between air karma (AK) and fluoroscopy time (FT) during TEVAR. (**b**) The relationship between AK and FT during EVAR. (**c**) The relationship between DAP and FT during TEVAR. (**d**) The relationship between DAP and FT during EVAR.

**Table 1 diagnostics-10-00846-t001:** The standard X-ray and contrast conditions used during endovascular treatment of aortic aneurysms in our hospital.

X-ray Conditions	Fluoroscopy	D A	D S A
Typical tube voltage (kV)	80	80	80
Typical tube current (mA)	50	400	125
Typical pulse widths (ms)	6	8	45
Pulse rate (p/s)	7.5	10	10
Additional filter	0.2mm Cu	0.3mm Cu	1.5mm Al
**Contrast Conditions**			
Injection method	—	Manual injection	Automatic injection
Flow rate (mL/sec)	—	≈ 0.2	10
Injection volume (mL)	—	≈ 5	20
Contrast agent	—	Iopamidol	Iopamidol
Iodine content (mg/mL)	—	370	370
		Cu: Copper, Al: Aluminum	

**Table 2 diagnostics-10-00846-t002:** Summary of data from the TEVAR and EVAR patients. The aneurysms were located in all four zones (thus zones 1–4) in TEVAR patients, but only one type of infrarenal abdominal aortic aneurysm was treated via EVAR.

Aneurysm Types of TEVAR	No. of Cases *n* = 84	Ratio (%)
Zone 0	27	32%
Zone 1	11	13%
Zone 2	15	18%
Zones 3–4	31	37%
**Aneurysm types of EVAR**	*n* = 172	
infrarenal AAA	172	100%

**Table 3 diagnostics-10-00846-t003:** A summary of our results: patient data and radiation doses.

	TEVAR	EVAR	*p*-Value (TEVAR vs. EVAR)
No. of Cases	84	172	
Age (years)	69.8 ± 10.8	73.8 ± 9.4	*p* < 0.01
Male-to-female ratio	72:12	147:25	n. s.
Height (cm)	163.2 ± 8.8	162.3 ± 7.9	n. s.
Weight (kg)	64.4 ± 10.5	62.1 ± 10.2	n. s.
FT (min.)	13.4 ± 7.1	23.2 ± 11.7	*p* < 0.01
AK (Gy)	0.92 ± 0.44	1.11 ± 0.54	*p* < 0.01
DAP (Gy·cm^2^)	323.7 ± 161.0	371.3 ± 186.0	*p* < 0.05
No. of acquisitions	18.7 ± 9.8	29.8 ± 11.0	*p* < 0.01
No. of stents	1.81 ± 0.65	2.49 ± 0.81	*p* < 0.01

**Table 4 diagnostics-10-00846-t004:** Overview of reports describing the endovascular treatment of thoracic aortic aneurysms (TEVAR) in HORSs.

Study of TEVAR	No. of Cases	Median FT (min.)	Median AK (Gy)	Median DAP (Gy·cm^2^)
Blaszak M A. [1], 2009	39	11.8 (13.4 ± 8.3)	0.87 (0.94 ± 0.56)	385.9 (452.3 ± 275.4)
Hertault A. [9], 2014	14	0.9 (—)	—	26.0 (—)
Panuccio G. [15], 2011	46	82.7 (111.3 ± 55.1)	6.30 (12.7 ± 4.4)	696.6 (1327.0 ± 469.9)
Zoli S. [16], 2012	48	12.3 (15.7 ± 11.4)	—	41.3 (56.7 ± 45.2)
Mohapatra A. [17], 2013	39	— (71.1)	— (7.0)	— (540.9)
Kirkwood M.L. [18], 2013	25	—	0.9 (—)	—
Sailer A.M. [28], 2015	11	— (9.7 ± 7.3)	—	— (62.0 ± 46.0)
Our study	84	12.1 (13.4 ± 7.1)	0.80 (0.92 ± 0.44)	299.7 (323.7 ± 161.0)

(Mean ± SD).

**Table 5 diagnostics-10-00846-t005:** Overview of reports describing the endovascular treatment of abdominal aortic aneurysms (EVAR) in HORSs.

Study of EVAR	No. of Cases	Median FT (min.)	Median AK (Gy)	Median DAP (Gy·cm^2^)
Blaszak M.A. [1], 2009	39	19.6 (22.8 ± 14.2)	1.11 (1.12 ± 0.70)	354.9 (380.9 ± 285.3)
Hertault A. [9], 2014	44	10.6 (—)	—	12.2 (—)
Van den Haak R.F. [10], 2015	18	— (13.6 ± 8.6)	1.26 (—)	224.4 (—)
	19	— (13.1 ± 6.1)	0.56 (—)	95.8 (—)
McNally M.M. [2], 2015	41	— (84.0 ± 36.0)	— (5.00 ± 0.28)	—
	31	— (55.0 ± 21.0)	— (2.30 ± 1.30)	—
Kirkwood M.L. [18], 2013	22	—	1.00 (—)	—
Majewska N. [19], 2011	92	13 (16.3)	0.63 (0.80)	456 (626)
Varu VN. [22], 2013	51	— (24.9 ± 12.4)	—	—
Tacher V. [26], 2013	9	— (82.0 ± 46.0)	—	— (1188.0 ± 1067.0)
	14	— (42.0 ± 22.0)	—	— (984.0 ± 581.0)
	14	— (80.0 ± 36.0)	—	— (656.0 ± 457.0)
Panuccio G. [27], 2015	150	— (89.0 ± 33.0)	—	— (60.7 ± 84.9)
Sailer A.M. [28], 2015	22	— (19.8 ± 8.4)	—	— (116.0 ± 122.0)
	11	— (49.1± 21.8)	—	— (217.0 ± 159.0)
**Our study**	172	20.5 (23.2 ± 11.7)	1.02 (1.11 ± 0.54)	345.3 (371.3 ± 186.0)

(Mean ± SD).

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
