# Peer review of "Hybrid Operating Room System for the Treatment of Thoracic and Abdominal Aortic Aneurysms: Evaluation of the Radiation Dose Received by Patients"

_diagnostics, 2020, doi:10.3390/diagnostics10100846_

Round 1

Reviewer 1 Report

Interesting prospective study concerning x-ray exposure during EVAR and TEVAR

in Hybrid operating room systems.

No mentions, athough cited in the manuscript,  on which of operator's behaviours can reduce exposition, i suggest to integrate with these information also citing some literature concerning this.   

Author Response

>Reviewer 1

>Comments and Suggestions for Authors

>Interesting prospective study concerning x-ray exposure during EVAR and TEVAR in Hybrid operating room systems.

>No mentions, athough cited in the manuscript, on which of operator's behaviours can reduce exposition, i suggest to integrate with these information also citing some literature concerning this.  

Thank you very much for you and your reviewer’s kind comments. Your comments have been helpful in allowing us to revise our manuscript.

We have been added the statement that

"Physician behavior can also reduce the radiation dose, such as use of the shortest fluoroscopy time possible [42]. The ICRP [43] also recommends that physicians, ‘‘Keep beam-on time to an absolute minimum—the golden rule for control of dose to patient.’’ "

(L 212)

The detailed review of this manuscript is appreciated and we have attempted to answer each of the questions raised. Thank you for your consideration of the revised version.

Sincerely,

Reviewer 2 Report

This is a very well written and organized manuscript on the radiation dose delivered by a number of intraoperative imaging techniques. I have very little to add.

General comments:

To be technical, air kerma (an acronym for Kinetic Energy Released per unit Mass) is a different quantity from radiation absorbed dose, even though they share a unit (Gy). Radiation dose to the patient (the quantity associated with increased risk) depends on the amount of x-rays that are backscattered from the patient, which in turn depends on a myriad of factors that are unknown to the imaging system, such as patient thickness, size of the projected field on the patient surface (which will depend on positioning, etc...). Since these are unknown, air kerma is used instead as it can be calculated from first principles using only the x-ray tube's technique settings (kV, mA, ms, pulse rate).

This may be a distinction that is only meaningful to physicists, but it may be more precise to refer to AK & DAP as "radiation dose indicators" rather than "radiation doses" per se.

Specific comments:

L78: I presume that the DAK, AK, FT reported are those recorded by the equipment themselves, using the manufacturer's calibration?

L90: Table 1 refers to "average" kV/mA/ms, etc. Typically these values are set for a given technique/anatomy, and do not change unless automatic brightness control (ABC) is used. Is that the case here? The values reported are rounded, which seems to indicate that they are nominal technique settings instead of being averaged over 80+ patients. If these are averaged values, how were they averaged by the investigators?

Fig1+Fig2: Is that a comic sans font for the x and y axis? Perhaps a typical font (e.g. Arial) would be more customary.

Fig1+Fig2: Given that the total beam-on time is what is the driver of radiation dose, I'm not sure why the number of acquisitions is particularly important adds as it shares a causal link with the fluoroscopic time, but is a less precise quantity (e.g. very short or very long acquisitions can distort the data vs total fluoroscopy time, whereas fluoroscopic time always increased radiation dose)

Tables 4+5: I believe the authors are using a Chōonpu (ー) instead of an em-dash (—). While they look similar in some fonts, in the one used in the table the Chōonpu  has clear curves. Was this intentional?

L182: "In such circumstances, the radiation dose is increased, because the X-rays need to travel for longer distances through tissue." The air-kerma is only increased if the kVp/mA is increased to compensate for the lack of penetration in the tissue. Is this the case here?

Author Response

>Reviewer 2

>Comments and Suggestions for Authors

>This is a very well written and organized manuscript on the radiation dose delivered by a number of intraoperative imaging techniques. I have very little to add.

Thank you very much for you and your reviewer’s kind comments. Your comments have been helpful in allowing us to revise our manuscript. We have attempted to address the questions raised by the referee according to the following:

>General comments:

>To be technical, air kerma (an acronym for Kinetic Energy Released per unit Mass) is a different quantity from radiation absorbed dose, even though they share a unit (Gy). Radiation dose to the patient (the quantity associated with increased risk) depends on the amount of x-rays that are backscattered from the patient, which in turn depends on a myriad of factors that are unknown to the imaging system, such as patient thickness, size of the projected field on the patient surface (which will depend on positioning, etc...). Since these are unknown, air kerma is used instead as it can be calculated from first principles using only the x-ray tube's technique settings (kV, mA, ms, pulse rate).

This may be a distinction that is only meaningful to physicists, but it may be more precise to refer to AK & DAP as "radiation dose indicators" rather than "radiation doses" per se.

"radiation doses" have been changed to “radiation dose indicators "

(L27, 62, 66, 226)

>Specific comments:

>L78: I presume that the DAK, AK, FT reported are those recorded by the equipment themselves, using the manufacturer's calibration?

We have been added the statement that

"The radiation dose indicators (DAP and AK) were calibrated by the manufacturer.”

(L82)

>L90: Table 1 refers to "average" kV/mA/ms, etc. Typically these values are set for a given technique/anatomy, and do not change unless automatic brightness control (ABC) is used. Is that the case here? The values reported are rounded, which seems to indicate that they are nominal technique settings instead of being averaged over 80+ patients. If these are averaged values, how were they averaged by the investigators?

"Average” have been changed to “Typical”

(Table 1)

>Fig1+Fig2: Is that a comic sans font for the x and y axis? Perhaps a typical font (e.g. Arial) would be more customary.

We have made the corrections requested. (Figs.2, 3, 4)

>Fig1+Fig2: Given that the total beam-on time is what is the driver of radiation dose, I'm not sure why the number of acquisitions is particularly important adds as it shares a causal link with the fluoroscopic time, but is a less precise quantity (e.g. very short or very long acquisitions can distort the data vs total fluoroscopy time, whereas fluoroscopic time always increased radiation dose)

We have been added the statement that

"Generally, correlations between the number of acquisitions and radiation dose indicators were low (Figs. 2, 3). Therefore, the number of acquisitions is not a useful parameter for radiation dose. "

(L205)

>Tables 4+5: I believe the authors are using a Chōonpu () instead of an em-dash (—). While they look similar in some fonts, in the one used in the table the Chōonpu has clear curves. Was this intentional?

We have made the corrections requested. (Tables 4, 5)

>L182: "In such circumstances, the radiation dose is increased, because the X-rays need to travel for longer distances through tissue." The air-kerma is only increased if the kVp/mA is increased to compensate for the lack of penetration in the tissue. Is this the case here?

We have been added the statement that

"(in other words, the kV/mA is increased to compensate for the lack of penetration in the tissue) "

(L191)

The detailed review of this manuscript is appreciated and we have attempted to answer each of the questions raised. Thank you for your consideration of the revised version.

Sincerely,
